# Intracardiac Thrombi in Preterm Infants—A Case Study and Review of the Literature

**DOI:** 10.3390/diagnostics13040764

**Published:** 2023-02-17

**Authors:** Ayala Gover, Dawod Sharif, Liat Yaniv, Arieh Riskin

**Affiliations:** 1Neonatal Intensive Care Unit, Bnai-Zion Medical Center, Haifa 3339419, Israel; 2Rappaport Faculty of Medicine, Technion, Israel Institute of Technology, Haifa 3200003, Israel; 3Department of Cardiology, Bnai-Zion Medical Center, Haifa 3339419, Israel

**Keywords:** intracardiac thrombus, preterm, intra-atrial thrombus, thrombolysis, thrombectomy

## Abstract

Intracardiac thrombi in preterm infants are not common but may lead to fatal outcomes. Predisposing and risk factors include small vessel size, hemodynamic instability, immaturity of the fibrinolytic system, indwelling central catheters and sepsis. In this paper, we present our own experience with a case of a catheter-related right atrial thrombus in a preterm infant, which was successfully treated with an aspiration thrombectomy. Then, we review the literature on intracardiac thrombosis in preterm infants: epidemiology, pathophysiology, clinical signs, echocardiographic diagnostic features and treatment options are discussed.

## 1. Introduction

Thrombotic events have been well described in preterm infants. Predisposing factors include small vessel size, hemodynamic instability causing changes in blood flow, immaturity of the fibrinolytic system and mechanical endothelial injury due to indwelling catheters [1]. Maternal risk factors such as diabetes and pre-eclampsia, and critical illness such as sepsis and asphyxia, further increase the risk of thromboembolic phenomena [2,3,4]. The incidence of symptomatic thrombosis in newborns (excluding stroke) greatly varies in different registries, from 2.4/1000 in infants admitted to the neonatal intensive care unit (NICU) to 5.1/100,000 and 0.7/100,000 of live births in other sources [2,4,5]. However, about 45–55% of these events occur in preterm infants in most epidemiologic reports [5].

Central lines, often essential for vascular access, are one of the most important contributing factors to thrombosis in the preterm population. In some studies, up to 90% of thrombotic events were found to be catheter-related [6], and about 9% of neonates with central catheters developed thrombosis [7]. Thrombosis can be venous or arterial [5] and may occur in various sites such as peripheral veins, portal vein, renal veins and cerebral arterial or sinovenous thrombosis [8].

Intracardiac thrombi in preterm infants are not common [9] but may lead to fatal outcomes [10,11]. We reviewed the literature on intracardiac thrombi, and present our own experience in this paper with a case of a catheter-related right atrial thrombus in a preterm infant successfully treated with an unusual procedure.

## 2. Case Presentation

A female preterm AGA (appropriate for gestational age; birth-weight 710 g) infant was born vaginally at 24 + 5/7 weeks of gestation after premature uterine contractions with cervical incompetence. Membranes ruptured 17 h prior to delivery. The mother received prenatal steroids, magnesium sulfate and antenatal antibiotics. The baby was vigorous at delivery and was immediately transferred to the neonatal intensive care unit (NICU) on nasal continuous positive airway pressure (CPAP) support. In the NICU, she was placed on non-invasive nasal intermittent positive pressure ventilation with an FiO_2_ of 0.21, and umbilical venous and arterial catheters were inserted for access, blood sampling and monitoring. Heparin 1 unit per ml normal saline, at a rate of 0.3 mL/h, was continuously run through the umbilical arterial catheter. However, heparin was neither run through the umbilical venous line nor added to the parenteral nutrition solutions given continuously via the umbilical venous line [12,13]. Abdominal and chest radiographs showed signs of mild respiratory distress syndrome (RDS) and appropriate positions of umbilical catheters’ tips (venous above the diaphragm at T8–9 thoracic spine level and arterial at T7–8). The baby was started on full parenteral nutrition, followed by the introduction of minimal enteral feedings of mother’s milk which were well tolerated and gradually increased. The infant had no polycythemia. Antibiotics were discontinued after 48 h, as blood cultures were negative. The baby continued to be hemodynamically stable with a stable respiratory status, did not need surfactant replacement therapy and required minimal non-invasive respiratory support with no supplemental oxygen. At 90 h of life, an echocardiogram was performed to assess for patent ductus arteriosus (PDA) due to active precordium and a systolic murmur. A two-dimensional echocardiographic study showed a widely open ductus arteriosus and a 9 mm long mobile thread-like echo-dense structure extending from the fossa ovalis and protruding to the inflow of the right ventricle, consistent with a thrombus (Figure 1, Appendix A). The umbilical venous catheter (UVC) was seen in the right atrium at the entrance of the inferior vena cava and the catheter tip was in close proximity, but not in contact with, the thrombus (Figure 2). Since the catheter tip was adjacent to the thrombus, an aspiration thrombectomy was attempted. Three 1.5 mL blood samples were drawn using 2 mL syringes through the umbilical vein 4F catheter (double lumen umbilical catheter, Ref. 1272.04; Vygon, Ecouen, France), with the catheter tip close to the thrombus. The aspiration was slow and gentle, with no added pressure required. In the second blood sample, a 9mm thrombus was detected (Figure 3). Echocardiography confirmed the disappearance of the right atrial thrombus (Appendix A). Since the thrombus was presumed to be catheter-related, the catheter was removed and traces of thrombi were noticed in the lumen. 

## 3. Discussion

### 3.1. Thrombus Formation in Preterm Infants

Immaturity of the coagulation system in preterm infants has been previously investigated. Coagulation factors are decreased in preterm infants compared to term infants or children, and there is evidence of reduced capacity for thrombin formation in this population [14]. However, reduced levels of coagulation inhibitors, a decreased ability for fibrinolysis and a higher activity of von Willebrand factor counterbalance this effect [14,15]. A prospective observational study in preterm infants performed coagulation assays on cord blood in infants born <30 weeks of gestation [16], and found prolonged PT (prothrombin time) and aPTT (activated partial thromboplastin time) compared to healthy term infants’ values, but no difference in plasma fibrinogen. In a small subset of patients, vitamin K-dependent procoagulant factors II, VII, IX and X were tested and found to be lower in preterm compared to term infants. However, levels of anticoagulant protein C, free protein S and antithrombin were also lower in the preterm group, so that the activity of both procoagulant and anticoagulant factors was reduced. 

In sick preterm infants, the fine hemostatic balance between pro- and anticoagulant pathways is often disrupted, leading to either thrombotic or bleeding complications. One study [17] obtained several hemostasis tests, including PT, aPTT, fibrinogen, thrombin-antithrombin complex, antithrombin III activity and fibrinolytic markers (d-dimer, plasminogen activator inhibitor, and plasminogen tissue activator), from ill preterm infants with RDS, sepsis or asphyxia and compared them to a group of healthy preterm infants. Their findings were consistent with the activation of the coagulation cascade and the fibrinolytic system in ill preterm infants at the initial phase of RDS and at the active phase of sepsis. It is believed that the limited reserve capacity of the immature hemostatic system in preterm infants renders them most vulnerable and at risk for thromboembolic events [14]. 

Additional predisposing factors for thrombosis in preterm infants include small vessel size, alterations in blood flow, central lines, polycythemia and the presence of maternal risk factors, such as diabetes and preeclampsia [2,3,4].

In a review of 26 studies on central line-related thrombosis in newborns [7], umbilical venous catheters and peripherally inserted central catheters were most commonly involved and the most frequent thrombus sites were the hepatic vein, right atrium and inferior vena cava. A venous catheter left in situ for more than 6 days, infusing blood products through a UVC and being small for gestational age were reported as risk factors for catheter-related thrombosis [3,7]. Interestingly, thrombophilia was not found to be associated with catheter-related thrombus in most reports [18,19,20,21]. 

### 3.2. Epidemiology of Intracardiac Thrombi in Preterm Infants

In a review of the pediatric literature, Yang et al. found that, out of 122 neonates and children with right atrial thrombi, 40.8% were preterm neonates [11]. However, the exact incidence of intracardiac thrombi in the preterm population is unknown. In one prospective study, screening echocardiograms were systematically performed in 49 preterm infants before and every 2–3 weeks after the insertion of central lines [22]; an incidence of 1.8% of right atrial thrombosis (one thrombus/56 central catheters) was recorded. In another study, screening echocardiograms were performed in the first four days of life on low-birth-weight preterm infants who had a central venous umbilical catheter confirmed to be in the inferior vena cava outlet or right atrium and an incidence of 5% (4 infants out of 76) of early intracardiac thrombosis was found [9]. Another prospective randomized controlled trial investigating UVC and peripherally inserted central catheters (PICC) complications in preterm infants born at ≤1250 g performed routine echocardiography at planned intervals; it was found that 24 out of 210 preterm infants (11%) developed an intracardiac thrombus over time [23]. The incidence was higher in small-for-gestational-age infants, and the median day of the first documentation of the thrombus was 11 days of age (range 5–30 days). Recently, a prospective study followed newborns with well-positioned UVCs on days 3, 7 and 14 after catheterization, and reported the incidence of right atrial thrombi to be 15% (6 thrombi/40 catheters) [24].

### 3.3. Pathophysiology and Risk Factors for Intracardiac Thrombi

In adults, intracardiac thrombi have been related to poorly contracting ventricles following an ischemic event, cardiomyopathy, valve stenosis and arrhythmia, leading to blood stasis [25,26]. Endocardial disease initiating thrombosis and migrating thrombi from other sites, such as deep peripheral veins, have also been described [26]. In newborns, ischemic heart disease, arrhythmia, valvular or endocardial disease are rarely involved, although left atrial thrombi have been anecdotally described following poor ventricular function due to septic shock or sustained supraventricular tachycardia [27,28].

Central lines are a major risk factor for intracardiac thrombi across all ages. The catheter material is a foreign body and is somewhat thrombogenic, and an intraluminal thrombus may extend from the line-tip to the atrium. Thrombi attached to the atrial wall that is separate from the catheter may be the result of direct endocardial injury due to the to-and-fro motion of the catheter tip in the right atrial cavity, or secondary to a jet lesion caused by the inflow of intravenous medication and hyperosmolar parenteral nutrition. Thrombi floating freely within the atrial cavity, whether catheter related or not, may migrate to other heart chambers [29] and lead to embolic phenomena [30,31]. 

Catheter malposition in the heart may increase the risk of thrombosis [11] although intracardiac thrombi have been reported in appropriately positioned catheters [32,33]. 

Sepsis is another major risk factor in preterm infants; thus, the combination of positive blood culture sepsis, an indwelling catheter and an intracardiac thrombus has been repeatedly described [9,27,34,35,36]. Staphylococcus aureus [27,34], coagulase-negative staphylococcus [37], Candida [29,35,38] and Group B streptococcus (GBS) [9] have all been isolated from blood cultures in these circumstances. In one case report of a preterm infant with GBS sepsis [9], an intracardiac thrombus, extending from the umbilical venous catheter tip, located in the atrium was already observed just 11 h after catheter insertion.

### 3.4. Clinical Signs of Intracardiac Thrombi

While catheter-related venous thrombosis may present with signs of catheter occlusion and malfunction, or local swelling and edema [7], intracardiac thrombi are often initially "silent" without specific cardiac clinical signs or symptoms [1,9,32,33]. A few cases were diagnosed when echocardiography was performed due to respiratory or general deterioration of the infant [20,33,39], and some cases presented with a new murmur [9,18,31,34], tachycardia [9] or bradycardia [11]. Rarely, hemodynamic instability was evident [29,34,37]. In one study, superior vena cava syndrome was reported in five extremely-low-birth-weight preterm infants with a large right atrial thrombus, leading to impaired cardiac preload and the need for vasoactive drugs [36]. Persistent thrombocytopenia was an unspecific sign of thrombosis [3,40].

### 3.5. Echocardiographic Diagnostic Features of Intracardiac Thrombi

The echocardiographic features of intracardiac thrombi are described in detail in many reports. The typical appearance is one of an echo-dense mass which can be irregular [9,41] or oval [18,35], floating [9] or attached to the atrial wall or septum [19]. It can be pedunculated and mobile [20,35] or sessile [23]. Significant variation in size was found across case reports, with the longest linear dimension measured to be between 3 mm and 33 mm [34,42], but in most case reports, thrombi were measured under 14 mm [9,19,20,23,33,34,35,37]. In one case of a term infant [20], a large, pedunculated well-defined mobile mass with a narrow stalk was observed, attached to the left atrial septum at the level of the fossa ovalis, and not interfering with mitral function or inflow. In this case, a myxoma was suspected based on sonographic appearance; however, the mass was resected and pathology analysis concluded it to be a thrombus. 

Most thrombi were located in the right atrium [9,18,33,35,39], although some cases were found in the left atrium [20], in the foramen ovale [43], at the tip of a catheter protruding to the left atrium through the foramen ovale [9,19,34] and in the left atrial appendage [27]. Some catheter-related thrombi extended from the inferior vena cava (IVC) or the superior vena cava (SVC) into the right atrium [3,23,34,40,42].

In some cases, a fibrin thread originating from the thrombus and extending into other heart chambers was seen. One case had a thrombus in the right atrium with a long fibrin thread extending through the foramen ovale into the left heart chambers and reaching as far as the aortic isthmus [9]. 

Many thrombi were noticed while the catheter was in place; however, others were diagnosed after the removal of the catheter [23].

While most cases did not affect heart function at diagnosis, some cases reported flow obstruction, paradoxical septal motion or valve dysfunction. Rapidly progressive heart failure was described in one preterm infant with sepsis and a catheter-related thrombus on the septal leaflet of the tricuspid valve, causing moderate tricuspid regurgitation [37]. In another case of a large thrombus filling most of the right atrium, the thrombus was seen partially entering the right ventricle during diastole [42]. One case report described a large thrombus in the right atrium partially obstructing the outflow tract to the right ventricle [40]. In another case, a small preterm infant developed episodes of bradycardia, respiratory distress, and poor peripheral perfusion due to a catheter-related thrombus, which initially extended from the SVC to the right atrium, continued through the tricuspid valve into the right ventricle, and subsequently migrated and completely blocked the right pulmonary artery, depriving perfusion of the right lung [29]. Another preterm infant with a central catheter tip lying close to the tricuspid valve had both a floating mass near the valve and a large mass in the pulmonary infundibulum that had prolapsed into the pulmonary artery during systole, causing a partial obstruction of the outflow tract [34]. 

In adults, the risk stratification of intracardiac thrombi is based on the morphologic characteristics and mobility of the thrombus [44]; however, no classification or risk stratification has been so far established in newborns. Yang et al. reviewed data from 122 infants, children and young adults with right atrial thrombi (the mean age was 3.58 years, with a range from 5 days to 20 years) [11] and classified 50 of the cases as high or low risk using available echocardiographic data. They considered high-risk features to be a thrombus that was sized above 20 mm and pedunculated, mobile or snake-shaped and mobile; however, the age and weight of these selected patients was not specified, so extrapolation from these data to the preterm population must be performed with caution due to the immense heart size differences between preterm infants and older children. Butler-O’Hara et al., in their study of catheter-related intracardiac thrombi in very low birth weight preterm infants, grouped thrombi by size as small (under 5 mm of longest linear dimension), moderate (5.1–10 mm), or large (above 10 mm), but considered the thrombus to be clinically significant only if there was evidence of threatened occlusion of a major vessel or crossing or blocking of a heart valve. In this study, six thrombi were moderate and the other 18 were small. Five out of six moderate size thrombi and 6 of 18 small thrombi were considered clinically significant, but none were obstructive [23]. 

### 3.6. Treatment Options for Intracardiac Thrombi in Preterm Infants

Treatment options for intracardiac thrombi in preterm infants include expectant management, anticoagulant treatment, thrombolytic treatment and thrombectomy. No randomized controlled trials are available so far to determine the best therapeutic approach [45]. In one study of 24 preterm infants with intracardiac thrombi, the thrombi resolved in all patients without any treatment, and upon echocardiographic follow-up, thrombi persisted for a median of 16.5 days (range: 5–279 days) [23]. In another case series of three preterm infants, spontaneous resolution of right atrial thrombi was observed in all of them by 6 months of age [40]. Complications of anticoagulant or thrombolytic treatment include intracranial, gastrointestinal or pulmonary bleeding and pulmonary embolism [1]. Nevertheless, in most reports, active treatment was undertaken. 

Anticoagulant treatment with unfractionated or low-molecular-weight heparin has been commonly used in the past and was occasionally reported to be successful, even though it is not a thrombolytic agent [27,33]. Levels of anti-Xa should be monitored for dosing corrections to achieve a therapeutic range [4,27]. 

Thrombolytic medications include streptokinase, urokinase and recombinant tissue plasminogen activator (rTPA) [1]. Streptokinase binds with plasminogen to form a complex that activates unbound plasminogen, catalyzing the conversion of plasminogen to plasmin, which actively degrades fibrin fibers in the clot. It has an unpredictable dose–response and a longer half-life than urokinase and has therefore been less used [1]. Urokinase has been successfully used in preterm infants with intracardiac thrombi [34,39,46], usually in doses between 1000 and 3000 U/kg/h [1], though higher doses have been documented [29]. Both urokinase and rTPA are plasminogen activators, but rTPA is thought to be more selective to clot fibrinolysis due to increased fibrin specificity and low affinity for circulating plasminogen. Additionally, rTPA has a shorter half-life and the hypocoagulative state may be rapidly reversed if needed. 

Recently, rTPA use has become more widespread [9,19,32,33,36]. In a review of the literature on rTPA treatment of intracardiac thrombi in preterm infants [45], data on 28 preterm infants <32 weeks were analyzed. A third of the patients received a loading dose (0.05–1mg/kg intravenously), and there was high variability in the continuous infusion dosing and duration (dosing ranged between 0.01 and 1.2mg/kg/hour, duration ranged between 1 and 13 days). A low dose (starting at 0.03 mg/kg/h) was used in 57% and the rest received high dose rTPA (0.5–0.6 mg/kg/h). Bleeding complications occurred in four infants (14%): intra-ventricular hemorrhage in three (severity unknown) and pulmonary hemorrhage in one. Three out of four of these occurred in patients treated with high-dose rTPA. Complete resolution of the clot was reported in 20 cases (71.4%), partial resolution in four (14%) and no resolution in two (7.1%). Eleven patients received prophylaxis with heparin after thrombolysis in variable dosages and durations. The authors concluded that thrombolysis should be used when conservative treatment with anticoagulation fails, and that treatment should be started at a low dose, gradually increasing if needed according to clinical response [45].

The infusion of thrombolytic agents directly to the thrombus via a catheter close to or embedded in the thrombus was performed in a few cases [1] in an attempt to obtain a local clot-specific response and decrease systemic effects. No hemorrhagic complications were reported, but in one case a systemic proteolytic state was made evident by the observation of decreased fibrinogen and elevated fibrin degradation products in serum [1]. 

Combined thrombolysis with both urokinase and rTPA has been used successfully [35], as has combined therapy with rTPA or urokinase and heparin [30,34,39], and rTPA and aspirin [41]. Some centers gave fresh frozen plasma prior to or with rTPA administration to ensure sufficient plasminogen levels [19,30]. 

Surgical thrombectomy may be considered in preterm infants with life-threatening intracardiac thrombi; however, there are no specific guidelines and an individualized approach considering risks and benefits should be employed in every case [47,48]. 

In our own case of an extremely premature infant, we employed an unusual approach by performing an echocardiographic-guided aspiration thrombectomy using the existing catheter found to be in close proximity to the clot; this approach was successful. We elected to try this approach because we were hesitant to use thrombolysis in such a young and fragile preterm infant in the first week of life, when bleeding complications are more common, and we considered the thrombus to have been recently formed, so it was presumed to be only loosely attached to the fossa ovalis. To our knowledge, such a procedure has not been reported previously. 

Timing of the removal of the catheter was inconsistent across studies. While some centers opted to replace the catheter immediately upon the diagnosis of intracardiac thrombi [40], others preferred to leave the catheter in place and correct its position or remove it after a few days of thrombolytic treatment or after the resolution of the thrombus [34,37,39,46]. Guidelines from the American College of Chest Physicians recommend the removal of the catheter in cases of right atrial thrombosis in children, but they do not include specific recommendations for preterm infants [49]. In one case report, the removal of an umbilical venous catheter with a large thrombus at its tip in the right atrium resulted in a tear of the distal part of the catheter and the catheter remnant was seen in the inferior vena cava [18].

### 3.7. Outcome and Follow up

Most reports document the disappearance of the thrombus over time [9,19,33,34,35,37,39], either with medical treatment or with conservative supportive care. In general, thrombolysis was shown to be successful within a few days [19,30,34], heparin use alone was successful within a few weeks [27] and spontaneous resolution with supportive care only was achieved after a few months [40]. Pulmonary or systemic embolism rarely occurred [11,30,31] and bleeding complications during treatment were uncommon [1,33]. Deaths were mostly unrelated to the treatment [9,32] and sometimes occurred prior to thrombolytic treatment [33]. After the regression of the thrombus, some experts recommend anticoagulation for 6 weeks to 6 months, depending on whether there are persistent thrombophilic factors [1]. 

Screening for thrombophilia remains controversial, and most experts do not recommend screening every newborn with an intracardiac thrombus but rather a targeted, individualized approach taking into account existing risk factors for thrombosis [2,5,50].

### 3.8. Prevention

Common-practice strategies to minimize catheter-related thrombosis include avoiding malpositioning, the restriction of catheter days and the use of continuous heparin infusion [1,51]. The effect of heparin was studied separately for peripherally inserted venous central lines and umbilical catheters. One prospective study randomized term and near-term newborns with UVCs to receive heparin infusion or a placebo through the UVC [52]. Most patients had the UVC inserted for exchange transfusion, congenital heart disease or respiratory failure. An unfractionated heparin infusion of 0.5 IU/mL in saline was given at a rate of 1 mL/h to the study group, and saline at the same rate was given to the control group. Serial ultrasound demonstrated no thrombi in the heparin group (*n* = 19), and one thrombus in the control group (*n* = 27); therefore, this finding was not statistically significant. To our knowledge, no other prospective studies of heparin use in umbilical venous catheters have been published so far. In a systematic Cochrane review of heparin use in umbilical arterial catheters [53], no effect of heparin on aortic thrombosis incidence detected by ultrasonography or a short aortogram was shown in the two studies examining this outcome. However, umbilical arterial lines differ from venous lines in their internal diameter of the catheter, as well as in the blood flow pattern in the vessel. Another Cochrane review assessed the effectiveness of heparin in the prevention of thrombosis in peripherally inserted central venous catheters in term and preterm newborns [54]. Three trials, with a total of 477 neonates, were included. No statistically significant differences were found between the heparin and no-heparin groups in the incidence of thrombosis, which was identified in one study by ultrasonography within 72 h of catheter removal [55] and in the other study by flushing the catheter after its removal and observing for clots [56]. In an observational study describing complications in 292 indwelling venous and arterial catheters in 130 patients (mostly preterm and term newborns), heparin continuous infusion was used prophylactically and no thrombosis was found [51]. However, in this study, thrombosis was only assessed clinically without the use of ultrasonography. Heparin bonding of catheters has been successful in preventing catheter-related thrombosis in animal studies, but no difference in the incidence of thrombosis assessed by ultrasonography was found between heparin-bonded and non-heparin-bonded catheters in children [57]. 

Early detection of intracardiac thrombi may improve treatment results and potentially decrease complications by identifying thrombi at the “silent”, non-obstructive stage. While it is currently not common practice to perform routine serial echocardiography on all newborns with central line catheters, other than when confirming their location, neonatologist-performed echocardiography has become an increasingly utilized tool across the world in the assessment of myocardial function, patent ductus arteriosus, persistent pulmonary hypertension and shock in preterm and term neonates, in the first few days of life, as well as later in their hospital stay [58,59,60,61]. The growing use of echocardiography as a point-of-care modality in NICUs may increase incidental early detection of intracardiac thrombi. One retrospective study examined routine echocardiography in preterm infants <29 weeks gestational age performed within the first 24 h of life for research purposes, and found that 30% of neonates had a 37% rate of various incidental findings on echocardiograms [62].

## 4. Conclusions

In conclusion, preterm infants are at risk for intracardiac thrombi, especially in the presence of indwelling catheters and systemic illness. Treatment approaches of supportive care, anticoagulation and thrombolysis have been described. Rarely, surgical thrombectomy is needed. In our case, the setting allowed for aspiration thrombectomy through an umbilical venous catheter which was successful. Screening for thrombophilia is controversial. 

## Figures and Tables

**Figure 1 diagnostics-13-00764-f001:**
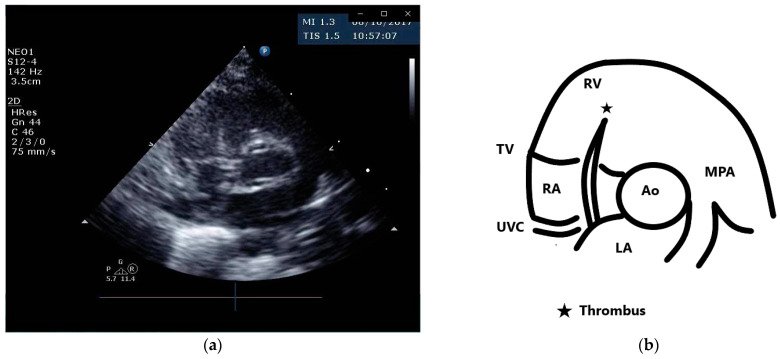
Parasternal short axis view and schematic drawing of the thrombus. (**a**) Short axis view showing the thrombus in the right atrium extending from the fossa ovalis and protruding to the inflow of the right ventricle; (**b**) schematic drawing of the echocardiographic view. *Ao,* aortic valve; *MPA,* main pulmonary artery; *LA,* left atrium; *RA,* right atrium; *RV,* right ventricle; *TV,* tricuspid valve; *UVC,* umbilical venous catheter.

**Figure 2 diagnostics-13-00764-f002:**
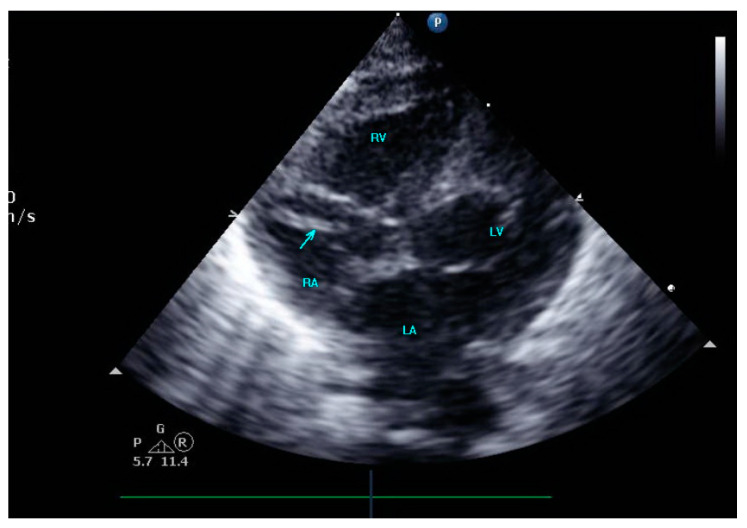
The umbilical venous catheter in the right atrium.

**Figure 3 diagnostics-13-00764-f003:**
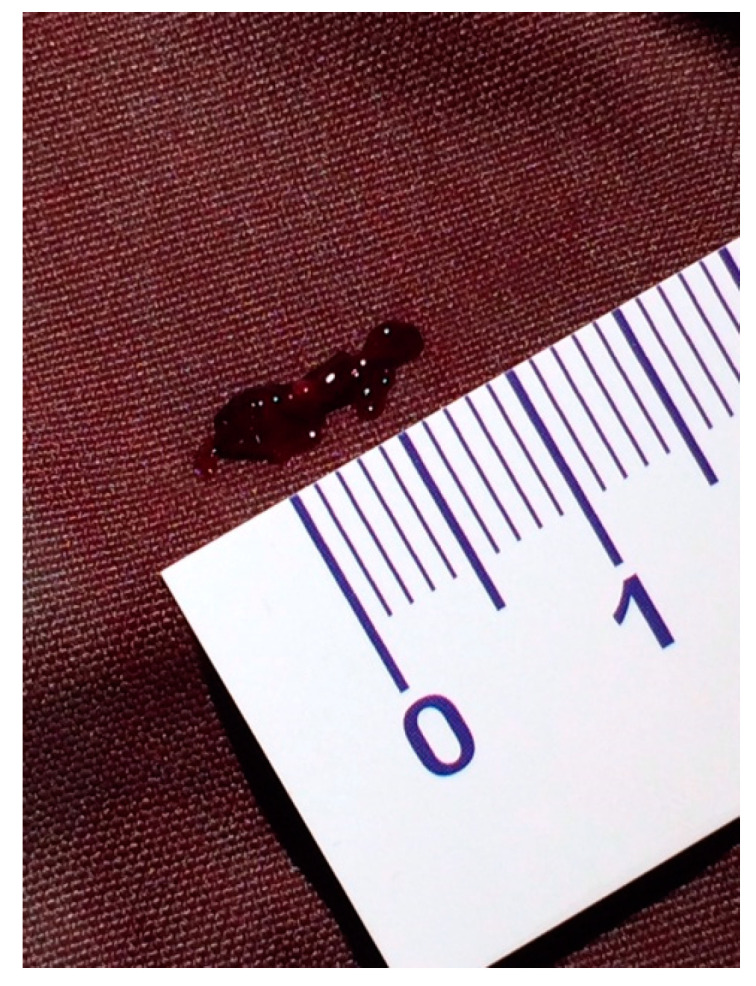
The aspirated thrombus.

## Data Availability

There is no extra data relevant to this case report and review beyond what is presented in the manuscript and Appendix A.

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
