# Peer review of "Intracardiac Thrombi in Preterm Infants—A Case Study and Review of the Literature"

_diagnostics, 2023, doi:10.3390/diagnostics13040764_

Round 1

Reviewer 1 Report

It is an interesting case report with educational value in the context of a well thought literature review.

Although you state that your treatment approach has not been attempted before, it would be appropriate to discuss your views further and explain why you approached the problem in the way you did (line 272-275).

Here are some comments and suggestions.

Abstract

Line 9: remove “a” and write “...but may lead to fatal outcome.”

Line 13: it would be more appropriate to say “...with aspiration thrombectomy. Then, we review the literature...”

Line 25: it would be more appropriate to write “...of live births [2, 4, 5]. However, about 45-55% of these events occur in preterm infants in most epidemiologic reports [5].

Line 29: replace with “about 9%”

Line 31: replace with “...may occur in various sites such as...”

Line 33: replace with “...but may lead to fatal outcome”

Discussion

Line 88: it would be more appropriate to write “...compared to term infants. However, levels of...”

Line 107: it would be more appropriate to say “A venous catheter left in situ for more than 6 days,...”

Line 176: it would be better to use a semicolon as follows “...on sonographic appearance; however, the mass...”

Line 189: replace with “flow obstruction”

Line 200-201: it would be more appropriate to say “Another preterm infant with a central catheter tip lying close to...”

Line 210: it would be better to use a semicolon as follows “...shaped and mobile; however, the age and...”

Line 222-223: it would ba more appropriate to say “No randomised controlled trials are available so far...”

Line 257: it would be more appropriate to say “The authors concluded that thrombolysis should...”

Author Response

Thank you for your helpful comments that will enhance the interpretation of this manuscript. We greatly appreciate the feedback and have attempted to make the appropriate adjustments.

Point 1 - It is an interesting case report with educational value in the context of a well thought literature review. Although you state that your treatment approach has not been attempted before, it would be appropriate to discuss your views further and explain why you approached the problem in the way you did (line 272-275).

Thank you for this important comment. We have now discussed the reasons for the decision to try aspiration thrombectomy in our patient (lines 284-288).

Point 2 - Here are some comments and suggestions.

Abstract

Line 9: remove “a” and write “...but may lead to fatal outcome.” Thank you for this comment. We have adjusted accordingly (line 9).

Line 13: it would be more appropriate to say “...with aspiration thrombectomy. Then, we review the literature...” Thank you for this comment. We have adjusted accordingly (line 13).

Line 25: it would be more appropriate to write “...of live births [2,4,5]. However, about 45-55% of these events occur in preterm infants in most epidemiologic reports [5]. Thank you for this comment. We have adjusted accordingly (lines 25-27).

Line 29: replace with “about 9%” Thank you for this comment. We have adjusted accordingly (line 30).

Line 31: replace with “...may occur in various sites such as...” Thank you for this comment. We have adjusted accordingly (line 32).

Line 33: replace with “...but may lead to fatal outcome” Thank you for this comment. We have adjusted accordingly (line 34).

Discussion

Line 88: it would be more appropriate to write “...compared to term infants. However, levels of...” Thank you for this comment. We have adjusted accordingly (line 98).

Line 107: it would be more appropriate to say “A venous catheter left in situ for more than 6 days,...” Thank you for this comment. We have adjusted accordingly (line 117).

Line 176: it would be better to use a semicolon as follows “...on sonographic appearance; however, the mass...” Thank you for this comment. We have adjusted accordingly (line 186).

Line 189: replace with “flow obstruction” Thank you for this comment. We have adjusted accordingly (lines 199-200).

Line 200-201: it would be more appropriate to say “Another preterm infant with a central catheter tip lying close to...” Thank you for this comment. We have adjusted accordingly (line 211).

Line 210: it would be better to use a semicolon as follows“...shaped and mobile; however, the age and...” Thank you for this comment. We have adjusted accordingly (line 220).

Line 222-223: it would more appropriate to say “No randomized controlled trials are available so far...” Thank you for this comment. We have adjusted accordingly (lines 232-233).

Line 257: it would be more appropriate to say “The authors concluded that thrombolysis should...” Thank you for this comment. We have adjusted accordingly (line 267).

Reviewer 2 Report

Please mention whether baby was SGA, any other risk factors like polycythaemia was there or not. Kindly describe the procedure in detail, what type of catheter was used and how much pressure was applied.

Was heparin infusion run through the UVC.

Author Response

Thank you for your helpful comments that will enhance the interpretation of this manuscript. We greatly appreciate the feedback and have attempted to make the appropriate adjustments

Point 1 - Please mention whether baby was SGA, any other risk factors like polycythemia was there or not. Kindly describe the procedure in detail, what type of catheter was used and how much pressure was applied.

Thank you for this important comment. We have now provided this information. The baby was AGA (line 41) and did not have polycythemia (line 57). The catheter was a 4Fr double lumen umbilical catheter (Vygon) (line 69-70). No extra pressure was applied during the aspirating procedure (lines 71).

Point 2 - Was heparin infusion run through the UVC.

Thank you for this important comment. We have now provided this information. A heparin infusion was not run through the UVC (lines 49-52). This is according to the ESPGHAN guidelines on pediatric parenteral nutrition (Section 9. Venous Access. Journal of Pediatric Gastroenterology and Nutrition 2005; 41:S57–S58; and Kolacek S, et al., ESPGHAN/ESPEN/ESPR guidelines on pediatric parenteral nutrition: Venous access (R 10.26), Clinical Nutrition 2018; 37:2379-2391).
